# The Effect of Amino Sugars on the Composition and Metabolism of a Microcosm Biofilm and the Cariogenic Potential against Teeth and Dental Materials

**DOI:** 10.3390/jfb13040223

**Published:** 2022-11-06

**Authors:** Lin Zeng, Alejandro Riveros Walker, Patricia dos Santos Calderon, Xinyi Xia, Fan Ren, Josephine F. Esquivel-Upshaw

**Affiliations:** 1Department of Oral Biology, University of Florida College of Dentistry, Gainesville, FL 32610, USA; 2Department of Dentistry, Federal University of Rio Grande do Norte, Natal 59056, RN, Brazil; 3Department of Chemical Engineering, University of Florida, Gainesville, FL 32611, USA; 4Department of Restorative Dental Sciences, Division of Prosthodontics, University of Florida, Gainesville, FL 32611, USA

**Keywords:** dental caries, amino sugars, GlcNAc, restorative surface, organic acids, microbial composition

## Abstract

Amino sugars N-acetylglucosamine (GlcNAc) and glucosamine (GlcN) are abundant sources of carbon and nitrogen in the oral cavity. The aim of this study was to investigate the effects of GlcNAc metabolism on the genomics and biochemistry of a saliva-derived microbial community, and on the surface integrity of human teeth and restorative surfaces. Pooled cell-containing saliva (CCS) was used to establish a microcosm biofilm in vitro in a biofilm medium (BM) containing 5 different carbohydrates. The microbial composition of each biofilm was analyzed by 16S rRNA amplicon sequencing, and the concentrations of eight organic acids were determined for selected sugars by targeted metabolomics. Meanwhile, extracted human teeth and polished titanium and ceramic disks were submerged in BM supplemented with 1% of glucose or GlcNAc, inoculated with CCS and *Streptococcus mutans* UA159, and incubated for 30 days. To mimic the effects of other microbial byproducts, the specimens were immersed in 10 mM hydrogen peroxide and 10 mM ammonium hydroxide for 30 days. The surface of each specimen was evaluated by profilometry for roughness (Ra) and imaged by scanning electron microscopy. The pH of the biofilm supernatant was significantly higher for the medium containing GlcNAc (*p* < 0.0001), and was higher in samples containing teeth than the two restorative disks for media containing the same sugar. For both teeth and titanium specimens, the samples treated with glucose-biofilm presented higher roughness values (Ra) than those with GlcNAc-biofilm and every other group. SEM images of the teeth and titanium disks largely supported the profilometry results, with glucose-biofilm samples demonstrating the largest deviation from the reference. For ceramic disks, slightly higher Ra values were obtained for the ammonia group. These findings provide the first direct evidence to support the ability of amino sugars to significantly reduce the cariogenic potential of oral biofilms by altering their biochemistry and bacterial composition. Additionally, amino sugar metabolism appears to be less detrimental to teeth and restorative surfaces than glucose metabolism.

## 1. Introduction

Dental caries or tooth decay is frequently accompanied by a change in the composition and biochemical activities of the microbiota due to organic acids released by bacteria. This changes microbiota from biofilms that are rich in health-associated commensals, to biofilms that contain substantially increased proportions of opportunistic, endogenous pathobionts [1,2]. Fermentation of dietary carbohydrates by these acid-resistant pathobionts drives down the biofilm pH as well as the abundance of commensals that help to resist colonization by these pathobionts [3,4,5]. *Streptococcus mutans* is one of the major pathogenic bacteria to colonize the tooth surface and these bacteria can be isolated from humans with or without dental cavities, suggesting that the oral environment of the host plays an important role in the virulence of the microbiome [6,7]. Consumption of carbohydrates, especially common dietary carbohydrates such as sucrose, glucose, and starch, has been inversely associated with the biodiversity of the supragingival microbiome, and positively associated with the abundance of several caries pathogens [8].

Amino sugars such as glucosamine (GlcN) and N-acetylglucosamine (GlcNAc) are abundant sources of carbon and nitrogen. They are available in the oral cavity and can be readily utilized by most bacteria of the microbiota. Provision of GlcN and GlcNAc to several oral commensal bacteria, in both planktonic and biofilm cultures, has been reported to enhance their fitness and antagonistic properties against *S. mutans* [9,10]. First, bacterial growth using amino sugars resulted in increased excretion by peroxigenic commensals of hydrogen peroxide (H_2_O_2_), which has a strong inhibitory effect on the growth and physiology of *S. mutans* [11]. Second, as a byproduct of amino sugar metabolism, ammonia neutralizes acids and can supply amino acid biogenesis. Amino sugar metabolism also promoted expression by commensals of arginine deiminase, which can act upon arginine for the generation of ATP and release of ammonia [12]. Furthermore, a recent transcriptomic study suggested the ability of amino sugars to modify bacterial central metabolism and reduce the production of lactic acid [13]. By increasing intracellular and environmental pH, the effects of amino sugars can greatly benefit the less acid-tolerant commensal species to potentially impede or even reverse the detrimental effects of aciduric pathogens at the microbiome level.

Surface integrity of teeth and restorative materials is essential in maintaining their normal functions, including adhesion, friction, hydrophobicity, biological response, and electrical and thermal transport properties. Dental ceramics are affected by stress, dynamic fatigue, and degradation of the surface due to exposure to acidic or alkaline pH, which may in turn influence their physical and mechanical properties [14,15]. ISO 6872 describes the standard for chemical durability in ceramic products by measuring weight change after immersion in pH 2 for 16 h. Studies have demonstrated that glass ceramics are subject to corrosion, not just from pH 2, but also from pH 7 and pH 10 [15]. Further studies demonstrated that an alternating pH environment [16], akin to intraoral changes brought about by dietary intake and the buffering capacity of saliva, is more detrimental to ceramic products than just a constant pH environment.

The formation of a titanium oxide layer on the surface makes titanium highly resistant to corrosion. However, the dissolution of this layer may occur in the presence of high fluoride concentrations, H_2_O_2_, and lactic acid [17]. In addition, the presence of *S. mutans* has been shown to also affect corrosion resistance [18]. Titanium has been the most successful dental implant material, owing to this metal’s unique characteristics of inertness and oxide formation that allows osseointegration around the peri-implant areas. However, corrosive titanium particles have been found in peri-implant sites that point to the possibility of corrosion contributing to the propagation of peri-implant disease [19,20,21,22].

Despite these apparent benefits of amino sugars in boosting the competitiveness of a handful of commensal bacteria, their impact at the biofilm level and on teeth and restorative surfaces remains unknown. In this study, we aimed to evaluate the ability of a saliva-derived microbial community, when supplied with amino sugars, to impact the surface integrity of three different dental specimens: titanium, ceramic, and human teeth.

## 2. Materials and Methods

### 2.1. Surface Sample Preparation

Fourteen samples of each material (ceramic, titanium, and human teeth) were used in this study. Ceramic discs (IPS e.max Press, Ivoclar Vivadent, 12 mm× 1.2 mm) were fabricated according to manufacturer recommendations and polished on both sides using 240, 400, and 600 grit silicon carbide abrasive paper (Carbimet, Buehler, Lake Bluff, IL, USA). Titanium rods at high purity (0.9999, TMS Titanium, Poway, CA, USA) were cut with Buehler Isomet 2000 into disks and polished to a grit size of 600 (Carbimet, Buehler, Lake Bluff, IL, USA) to 10 mm × 2 mm. After preparation, ceramic and titanium samples were washed three times in ethyl alcohol and dried with compressed air.

Collection of teeth was approved under the University of Florida Institutional Review Board (Protocol No: IRB201602423). Collection jars were placed in the oral surgery department with instructions to collect intact extracted anterior and posterior teeth. Contents of the jars were collected weekly. The teeth were immersed and sanitized in sodium hypochlorite (10% concentration). Anterior teeth were selected for this experiment in anticipation of the profilometry examination, which yielded more accurate results on flatter surfaces. Teeth were used without any previous preparation. For the experiment, teeth were washed in distilled water and then in ethyl alcohol, and then dried with compressed air.

### 2.2. Formation of Saliva-Derived Mmicrocosm Biofilm, Taxonomic Analysis, and Targeted Metabolomics

Cell-containing saliva (CCS) samples were collected from four adult volunteers, who were nonsmokers and had not taken antibiotics for at least three months (Protocol No: IRB201500497). Prior to saliva collection volunteers were told to avoid eating, drinking, and brushing for at least 2 hrs. Approximately 20 mL of unstimulated saliva was collected from each participant by expectorating into a sterile plastic tube on ice. Equal volumes of untreated whole saliva from each volunteer were pooled. Glycerol was added to a final concentration of 25%, and the CCS samples were stored in aliquots at −80 °C.

To form biofilms, aliquots of CCS were used to inoculate a biofilm medium (BM) composed of potassium phosphate buffer (72 mM), various vitamins, amino acids, and essential metal salts [23]. To mimic the oral environment, 2 mM of sucrose was added to all media, in addition to 18 mM of glucose (Glc), GlcN-HCl (CAS#66-84-2), GlcNAc (CAS#7512-17-6), or 9 mM of sucrose or lactose (MilliporeSigma, Rockville, MD, USA) [23]. The cultures were maintained in a 6-well plate containing a sterile glass coverslip and incubated at 37 °C in an ambient atmosphere maintained with 5% CO_2_ for 48 h, with a medium refreshment at the 24 h point. After a final medium refreshment, the supernatant and the biofilm mass were harvested by centrifugation and stored at −80 °C.

Chromosomal DNA was extracted from the saliva and the culture pellets using Promega Wizard Chromosomal DNA extraction kit (Promega) and subjected to PCR amplification using a pair of primers (QIAseq 16S/ITS Panels, Qiagen) designed to target the V2–V3 variable region of the bacterial 16S ribosomal RNA gene. The purified 16S rRNA amplicon was subsequently sequenced on the Illumina MiSeq platform at the University of Florida ICBR Next-Gen sequencing core laboratory. Taxonomic analysis was performed in the HiperGator3 cluster computer at the University of Florida. After a quality check with FASTQC [24], the transcripts were assembled using Megahit [25], and the resulting contigs were mapped with BLASTN [26] against an updated local nucleotide blast (NT) database. Next, we created a local BLASTN database to which the raw sequences were mapped and classified and annotated by taxID from NCBI taxonomy. Sample counts were merged together by taxID and moved to R statistical software [27] for further analysis and plotting with the package RAM [28].

Based on the microbial composition of each biofilm and our prior research on amino sugars and individual bacterial species [10,13], bacterial cells from CCS biofilms formed with glucose or GlcNAc were selected for metabolomic analysis using LC-MS targeting 8 organic acids, namely lactate, pyruvate, succinate, 3-hydroxybutyrate, α-ketoglutarate, malate, citrate, and fumarate. Bacterial cell pellets were shipped to the service provider on dry ice and stored at −80 °C. At the time of preparation, the frozen cell pellets were lyophilized to dryness, weighed, and homogenized in 500 μL of 50/50 acetonitrile/0.3% formic acid using a bead-based homogenizing system (Precellys). The resulting lysate was aliquoted for the metabolite assays and stored at −80 °C. Each sample was represented by 3 biological replicates. Statistical analysis was performed by the service provider at the Advent Health Translational Research Institute (Orlando, FL, USA).

### 2.3. Treatment of Tooth and Restorative Surfaces with Biofilm Cultures

*S. mutans* strain UA159 was inoculated into brain heart infusion (BHI; Difco Laboratories, Detroit, MI, USA) medium and cultivated overnight at 37 °C in an ambient atmosphere maintained with 5% CO_2_. After washing once using BM base without any carbohydrates, the bacterial suspension was kept on ice until further use.

To study the effects of saliva-derived microbial community and amino sugar (GlcNAc) on the integrity of various surfaces, extracted human teeth and polished (600 grits) titanium and ceramic disks were sterilized by autoclave for 20 min, then submerged in 2 mL (teeth) or 1 mL (disks) of BM medium containing 2% (*v*/*v*) of CCS, 1 × 10^6^ UA159 cells (based on optical density), 2 mM of sucrose and 1% of additional carbohydrate as specified. Six samples of each specimen were kept in a 24-well plate, 3 in 1% of glucose, and the other 3 in 1% of GlcNAc. The teeth were placed with the entire crown submerged and the vestibular surface facing up. The plate was kept at 37 °C in an incubator and maintained with 5% CO_2_. The media were refreshed once a day throughout the 30-day experiment. At the end of the experiment, the biofilms were scraped off, and treated teeth and disks were washed 3 times in ethyl alcohol and dried with compressed air.

### 2.4. H_2_O_2_ and NH_4_OH Immersion

Previous studies [9,10,13] showed that oral streptococci released increased amounts of H_2_O_2_ and ammonia after GlcNAc metabolism under similar conditions. To evaluate the effect of these products individually, 6 samples of each group were used. Each sample was placed in a 15 mL polypropylene Falcon conical tube filled with 10 mL of 10 mM H_2_O_2_ solution or 10 mM NH_4_OH solution for 30 days at room temperature. Three samples of each group were placed in each solution, and the solution was changed every 2 days. At the end of the experiment, samples were washed 3 times in ethyl alcohol and dried with compressed air. The pH of each solution was measured prior to dilution. The pH for 30% H_2_O_2_ was 3.5 and the pH for 5% NH_4_OH was 10.5. After dilution to 10 mM, both solutions registered a pH of 6.6, being almost neutral.

### 2.5. Profilometry

After 30 days of incubation, two samples of each specimen from each experiment, and one reference sample of each specimen, were subjected to surface roughness measurements using an optical noncontact profilometer (Bruker 3D Contour GT-I. Bruker, Tucson, AZ, USA). Values were obtained regarding arithmetical roughness (Ra). The sample surfaces were evaluated using a 20× magnification lens on a 0.15 × 0.12 mm field of view at a scan speed of 1×. Each sample was evaluated at three randomly selected areas, and the roughness value was the average of this measure for each sample. The quantitative data were reported as the averages ± standard deviations.

### 2.6. Scanning Electron Microscopy

After the 30-day experiment, one sample of each specimen from each treatment group and one reference sample of each material were assessed under scanning electron microscopy (FEI NOVA NanoSEM 430, FEI Company, Hillsboro, OR, USA) to qualitatively evaluate surface topographical changes. Samples were sputter-coated with carbon to reduce charging effects during SEM analysis (10–15 mA, under a vacuum of 130 mTorr). The SEM was operated at 10 kV, spot 3.5.

## 3. Results

### 3.1. Amino Sugars Enhanced Biodiversity of Saliva-Derived Microcosm Biofilm

Previous research on amino sugars has shown their ability to impact competition among oral streptococci. To evaluate their effects on microbial ecology on a larger scale, an ex vivo, CCS biofilm model was adopted using pooled saliva samples from healthy donors and 5 carbohydrates commonly available to oral microbiota, including GlcN and GlcNAc. 16S rRNA amplicon sequencing allowed for distinction in the bacterial makeup at the genus and species levels. Overall, the study identified 35–37 major taxa in each biofilm known to populate the supragingival plaque [29]. Figure 1 illustrates the relative abundance of some of the most prominent species as a function of the carbohydrate source. Compared to the original saliva sample, cultivation under our laboratory condition greatly enhanced the proportion of *Streptococcus salivarius* while reducing that of other species such as *Provetella melaninogenica* and *Neisseria subflava*. Among 5 carbohydrates, glucose, lactose, and sucrose produced similar patterns of taxonomic content, with very few streptococcal species other than *S. salivarius*, despite notable variations among replicates of glucose and lactose groups. Conversely, in CCS biofilms formed with GlcN and especially GlcNAc, significant expansion by the species of *S. mitis, S. oralis, S. cristatus, S. sanguinis,* and *Haemophilus parainfluenzae* was evident, along with the reduction in the abundance of *S. salivarius*. These results are consistent with our previous findings suggesting the beneficial effects of amino sugars, GlcNAc in particular, to these members of the mitis group streptococci [10].

### 3.2. Growth on GlcNAc Altered Central Metabolism and Acid Production

Considering the effect of GlcNAc on the biodiversity of the CCS biofilms, we collected bacterial cultures from 2-day-old biofilms formed on glucose or GlcNAc and subjected them to targeted metabolomic analysis. Measurements of 8 organic acids that are commonly produced during bacterial metabolism and the citric acid cycle are presented in Figure 2. Among all 8 organic acids measured here, lactate had the highest concentration, followed by pyruvate. Acetate was not included in the package offered by the service. Consistent with the fact most of the oral streptococci lack a complete citric acid cycle, several acids were detected in very low levels, namely fumarate, malate, α-ketoglutarate, and citrate. 3-hydroxybutyrate is a substrate for the synthesis of polyhydroxybutyrate, which serves as an energy reserve in certain bacteria including lactic acid bacteria [30,31]. Compared to glucose-grown samples, biofilms formed on GlcNAc yielded significantly lower levels of lactate as well as pyruvate. Our previous research in *S. mutans* has indicated similarly lower lactic acid production in cells growing on GlcNAc; however, the same cells also produced greater amounts of pyruvate compared to glucose conditions [13]. GlcNAc-grown biofilms also produced significantly lower amounts of succinate, fumarate, and α-ketoglutarate, but slightly higher levels of malate. Together, these results indicated a metabolic shift in the microcosm biofilm caused by GlcNAc, including a reduction in the production of lactic acid, a finding that was consistent with the expansion of mitis group streptococci which are considered beneficial to dental health.

### 3.3. Growth on GlcNAc Reduced the Ability of the Biofilm to Erode Teeth and Restorative Surfaces

Human teeth and two restorative surfaces, titanium and ceramic, were treated for 30 days with CCS biofilms in the presence of either glucose or GlcNAc. pH of the spent culture media was monitored periodically, which showed similar trends throughout the 30-day period. A representative set of pH values are presented in Figure 3. Biofilm cultivated with GlcNAc consistently produced higher pH than those cultivated with glucose, which was consistent with the finding that GlcNAc resulted in reduced lactate and pyruvate production (Figure 2). Meanwhile, biofilms cultured together with human teeth had higher pH values than those cultured with two restorative materials, regardless of the sugar included in the medium. This result could be interpreted as the leaching of substances out of the teeth, e.g., calcium phosphate, which acted as a buffering reagent that impeded acidification of the medium. To assess the effects of metabolites other than acids, these three surfaces were similarly treated with 10 mM of NH_4_OH or 10 mM H_2_O_2_ for 30 days to assess the effect of these byproducts on the surfaces of teeth and restorative materials.

Subsequently, profilometry was performed on these 3 surfaces to assess the impact of biofilms and chemical erosion relative to untreated references. Overall measurements of surface roughness (Ra values in Figure 4, and 3D images in Figure 5) showed the highest relative change of Ra values on ceramic disks, followed by teeth and titanium disks (Table 1). Regarding the carbohydrates used in biofilm media, glucose resulted in a significantly greater increase in Ra values than GlcNAc, especially on ceramic, which had an almost 100% increase in relative roughness (Table 1), followed by teeth and titanium that each showed about a 50% increase in Ra value. Note that these changes in Ra values confirmed the above result of higher lactic acid production and lower pH values in biofilms formed with glucose than with GlcNAc. With the exception of ceramic disks treated with NH_4_OH, the glucose-treated samples exhibited the highest increase in relative Ra values.

Treatment with NH_4_OH or H_2_O_2_ generally showed little to no effect on the Ra values for teeth or titanium. However, the ceramic disks had a relative increase of 154% Ra value for the NH_4_OH group, suggesting a particular sensitivity to alkali.

SEM images in Figure 6 illustrate representative surface topology of each sample derived from their profilometry measurements. It is worth noting that the scale used for the ceramic specimens is in nm, and that for titanium and teeth is in µm. Ceramic disks appear to be less rough in these images compared with the profilometry results. This just shows that the ceramic samples are relatively smoother than the titanium or the teeth, even when compared to the reference samples.

## 4. Discussion

Excessive consumption of carbohydrates is a major risk factor for caries development since their fermentation by lactic acid bacteria invariably results in biofilm acidification and, in longer terms, loss of bacterial homeostasis. During fasting periods of the host, however, the oral microbiome turns to endogenous glycans for sustenance. GlcNAc is naturally present in the oral cavity, as is its deacetylated derivative GlcN. It is estimated that as much as 80% of the total mass of a salivary mucin molecule can be composed of carbohydrates, including GlcNAc [32]. Previous research has demonstrated the effects of GlcNAc in modifying bacterial central metabolism and in promoting the competitiveness of several abundant commensal bacteria against the major caries pathogen *S. mutans* [9,10,13]. The present study was designed to test the possibility of extrapolating these anti-*S. mutans* potentials of amino sugars to the level of dental biofilm. To explore the contribution of amino sugars to the physiology and ecology of the oral microbiome, we analyzed a saliva-derived biofilm model and assessed its impact on the integrity of both human teeth and two restorative materials. Compared to glucose which represented dietary sugars, use of GlcNAc resulted in enhanced bacterial biodiversity and less lactic acid production by the biofilm. Prolonged treatment (30 days) of dental materials with such a biofilm demonstrated a significantly smaller impact on the surface integrity of teeth, titanium, and ceramic disks (Table 1). At the same time, GlcNAc-grown biofilms consistently allowed for higher pH than glucose-grown biofilms.

Compared to glucose, the metabolism of GlcN and GlcNAc comes with additional metabolic byproducts: ammonia as a product of deamination of GlcN-6-P, and often, with peroxigenic oral streptococci, increased amounts of H_2_O_2_. Esquivel-Upshaw et al. [15] found significantly more pitting and loss of surface structure of ceramic specimens immersed in a pH 10 buffer solution when compared to the specimens immersed in a pH 2 buffer, suggesting that an alkaline environment can be more detrimental to the ceramic surface. On the other hand, dissolution of the titanium oxide layer, which is responsible for titanium’s resistance to corrosion, may occur in the presence of high H_2_O_2_ and lactic acid [17,33,34]. To test the effects of these additional metabolites, we subjected the test materials to 10 mM of H_2_O_2_ or 10 mM NH_4_OH for the same period of time, a concentration that is close to but likely higher than what is expected of both metabolites in actual bacterial cultures [10,12,35]. Ceramic disks underwent a 153.79% increase in Ra value relative to the reference with NH_4_OH treatment. While the pH for the diluted solution was about 6.6, the hydroxyl group is what is proving to be detrimental to the ceramic structure. Esquivel-Upshaw et al. [16] confirmed that an ionic exchange mechanism occurs with acidic or predominantly hydrogen-rich solutions. The ionic exchange occurs between the H ions in solution and the network modifiers in the ceramic such as Na^+^ and K^+^. This leads to the formation of a corrosion-resistant layer and does not contribute to excessive surface degradation. In contrast, hydroxyl-rich solutions cleave the Si-O bonds and result in an attack of the network formers by the -OH group. This results in surface degradation and loss of ions in solution. Ammonia released by bacteria during GlcNAc metabolism is likely neutralized by acids and/or assimilated for biosynthetic processes. Taken together with the effects of the biofilms, we can deduce that the primary effect of GlcNAc in reducing the cariogenic potential of the biofilm was by maintaining an elevated pH of the medium, which was likely realized by releasing ammonia and by reducing the production of organic acids such as lactic acid.

Biological environments are highly aggressive for metallic implants. Based on their findings, Souza et al. [18] reported that *S. mutans* colonies on titanium surfaces negatively affected the corrosion resistance, based on titanium passive films’ polarization resistance. The authors concluded that acidic substances released by *S. mutans* metabolism can cause titanium corrosion when exposed to high sucrose concentrations for a long period of time. Souza et al. [18] reported pitting corrosion in titanium surfaces with 10% H_2_O_2_, but not at 0.1%. However, the effects of H_2_O_2_ on enamel are unclear, according to Pinto et al. [36], as the group using 35% of H_2_O_2_ presented significantly higher Ra values than the control group, whereas Carvalho et al. [37] did not find changes in enamel surface roughness after treatment with 9.5% or 38% H_2_O_2_, and Faraoni-Romano et al. [38] showed significant decreases in Ra values following bleaching with 7.5%, 18%, and 38% H_2_O_2_.

Biofilm accumulation can be increased by changes in the topography and roughness of restorative materials and teeth, which is associated with increased risk for periodontal diseases and caries, in the least affects aesthetics of teeth and ceramics. In the early stages of biofilm accumulation, bacteria accumulate in pits and grooves and then spread over the surface of the tooth [39]. It has been suggested that polishing should result in a final surface roughness below the Ra threshold of 0.2 µm for the clinical acceptability of restorative materials [40]. As such, when polishing enamel or titanium, extra care must be taken in order to reduce the likelihood of biofilm accumulation due to an increase in roughness. Furthermore, other physicochemical properties of bacteria and material surfaces can also affect bacterial adhesion, including surface energy, chemical composition, surface charge, and hydrophobicity [41], and further studies should be conducted to evaluate these additional material properties in the context of amino sugar metabolism by *S. mutans* and commensals. In addition to the apparent benefits of amino sugars metabolism in boosting the competitiveness of commensal bacteria, the lower impact on teeth and restorative surfaces is positive to avoid corrosion in these surfaces.

## 5. Conclusions

Our findings provide support for the ability of amino sugars to reduce the cariogenic potential of oral biofilms by changing their biochemistry and bacterial composition. Compared to glucose, the metabolism of GlcNAc by an ex vivo microcosm biofilm showed a smaller impact on the surface integrity of teeth and restorative surfaces. When used independently, ammonia is the only product of this process that seemed to negatively impact ceramic surfaces.

## Figures and Tables

**Figure 1 jfb-13-00223-f001:**
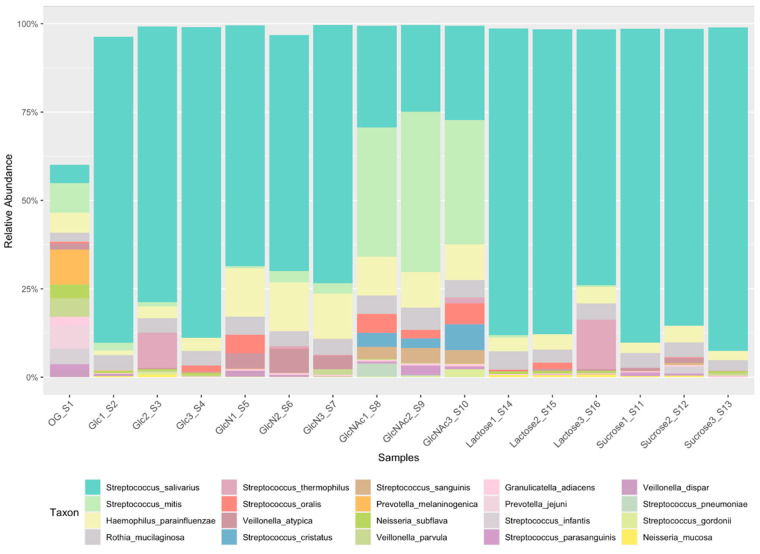
Relative abundance of most common species identified in CCS biofilms formed under different carbohydrates.

**Figure 2 jfb-13-00223-f002:**
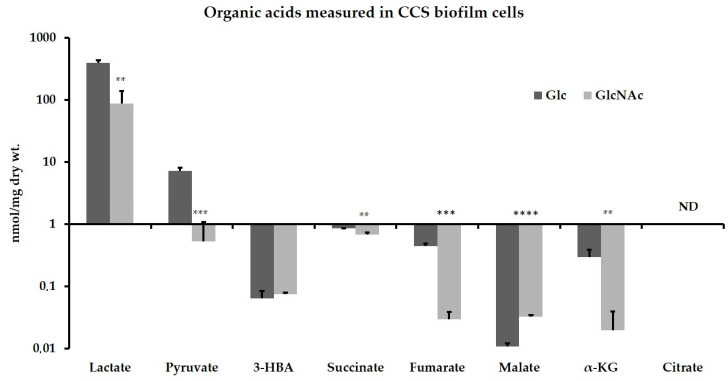
Measurements of organic acids in CCS biofilms. 2-day CCS biofilm cultures were prepared with BM with 2 mM sucrose and 1% of glucose (Glc) or GlcNAc. Eight hours after refreshing media, total biomass was harvested for targeted metabolomic analysis of 8 organic acids. ND, not detected. Asterisks indicate statistical significance analyzed by Student’s *t*-test (**, <0.01; ***, <0.001; ****, <0.0001). 3-HBA, 3-hydroxybutyric acid; α-KG, α-ketoglutarate.

**Figure 3 jfb-13-00223-f003:**
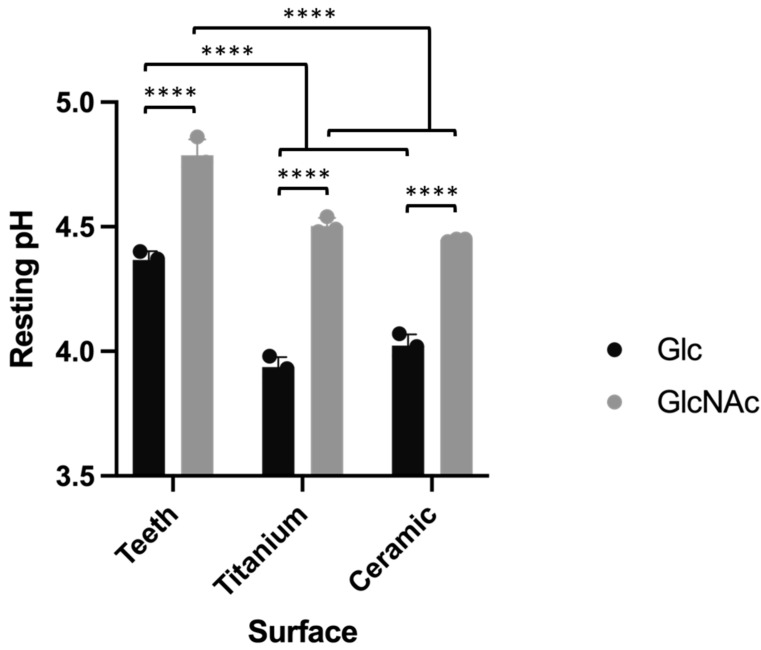
Representative pH measurements during the 30-day experiment with daily medium refreshment. Result of each treatment is the average and standard deviation (error bar) of 3 individual repeats. Asterisks indicate statistical significance between sugars analyzed by Student’s *t*-test (****, <0.0001).

**Figure 4 jfb-13-00223-f004:**
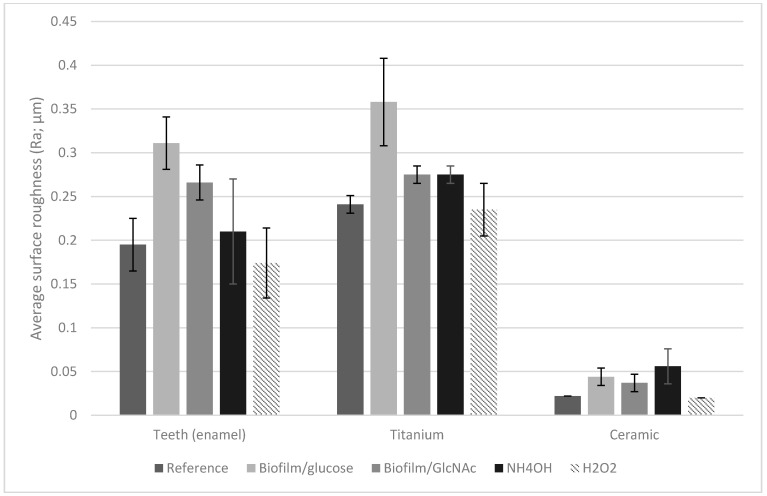
Average surface roughness ± SD for each specimen in each group.

**Figure 5 jfb-13-00223-f005:**
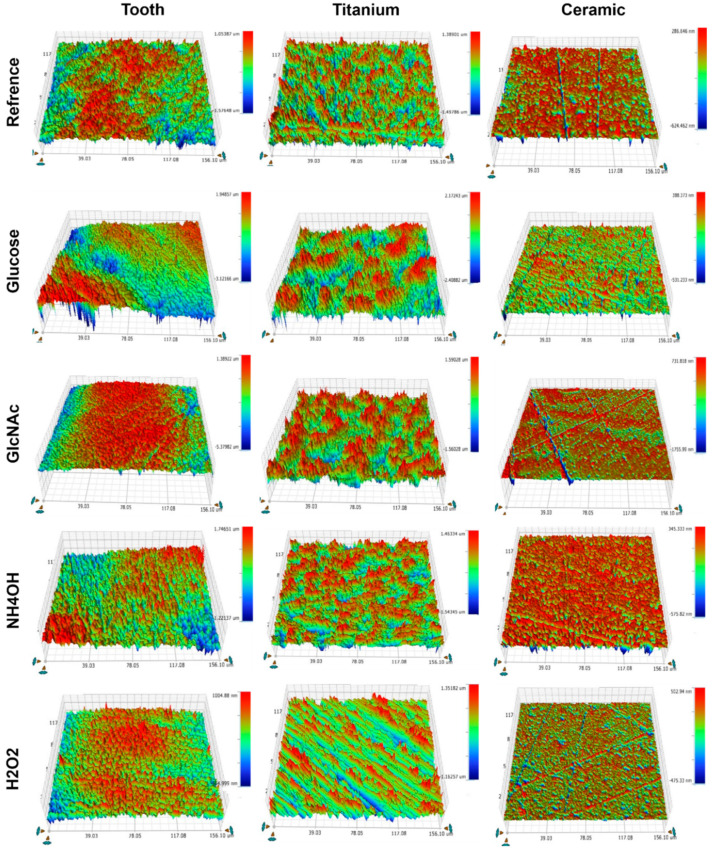
Profilometry 3D images showing different topographic pattern for different specimens.

**Figure 6 jfb-13-00223-f006:**
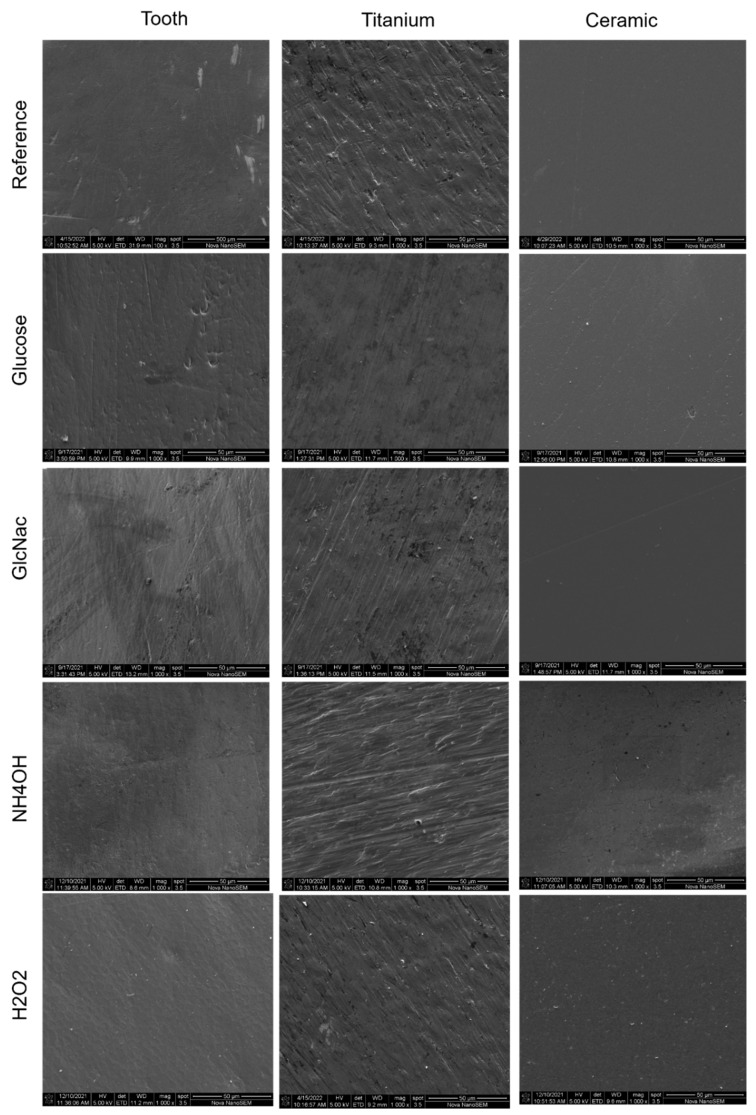
SEM images showing the surface of the specimens.

**Table 1 jfb-13-00223-t001:** Percent change in Ra values relative to the reference after treatment.

	Glucose (%)	GlcNAc (%)	NH_4_OH (%)	H_2_O_2_ (%)
Tooth	59.66	36.50	7.44	10.85
Titanium	48.41	14.26	13.97	2.63
Ceramic	99.24	66.67	153.79	7.58

## Data Availability

The high-throughput sequencing data from this study have been deposited with the Sequence Read Archive (SRA) and assigned accession number PRJNA862134.

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
