# Peer review of "The Effect of Amino Sugars on the Composition and Metabolism of a Microcosm Biofilm and the Cariogenic Potential against Teeth and Dental Materials"

_jfb, 2022, doi:10.3390/jfb13040223_

Round 1
Reviewer 1 Report
My comments and inquiries are noted below.
1. Re: The chemicals used this study was not identified.
Was Glucosamine glucosamine sulfate or Glucosamine Hydrochloride used? CAS number, the supplier’s name and location are also needed.
2. Line 111. The teeth were immersed and sanitized in sodium hypochlorite (10% concentration).
How long the teeth immersed in sodium hypochlorite (10% concentration) until used? Sodium hypochlorite is a strong oxidizer, and oxidation reactions are corrosive. The teeth were chemically damaged.
3. Line 129. 18 mM of glucose, GlcN, GlcNAc, or 9 mM of sucrose or lactose.
I think that the concentration of these addiction was 18mM for mono- and 9mM for disaccharide sugar. Does 18mM or 6 mM of saccharide sugar mimic the oral environment? Please provide related reference(s).
4. Line 149
Method of measurement of organic acids using LC-MS was not explained in Materials and methods, but in the caption in Figure 2. Please explain its detail in Materials and methods.
5. Line 163. 2 mM of sucrose and 1% of additional carbohydrate as specified. Six samples of each specimen were kept in a 24-well plate, 3 in 1% of glucose, the other 3 in 1% of GlcNAc.
i) GlcN was not used in organic acids measurement and evaluation erode teeth and restorative surfaces. Are there any difference in the effect between GlcNAc and GlcN (if used)?
ii) Figure2 and 3 shows Glc and GlcNAc. Abbreviation of Glc is not defined in the manuscript.
6. Figure 1 indicated a metabolic shift in the microcosm biofilm caused by GlcNAc, but not by GlcN, Why?
7. For Fig. 3 and 4, statistical analyzed methods were not fully described.
How about normal distribution and analysis of variance?
8. Line 194. Samples were sputter-coated with carbon to reduce charging effects during SEM analysis
If not EDS elemental analysis, carbon is not the best choice for coating. Gold or platinum is a better material than carbon.
.
9. Line 280. Figure 5. and Table 1.
i) The groups are placed in the order of tooth, titanium and ceramic in Figure 5, but in the order of tooth, ceramic and titanium in Table 1. Please arrange them consistently.
ii) In Figure 5, especially ceramic samples, scratches and streaks from polishing are visible. The use of filtering to remove these artifacts may be recommended. 
Author Response
We have carefully reviewed the comments and have revised the manuscript accordingly. Our responses are given in a point-by-point manner below. Changes to the manuscript are shown in track changes.1. Re: The chemicals used this study was not identified.
Was Glucosamine glucosamine sulfate or Glucosamine Hydrochloride used? CAS number, the supplier’s name and location are also needed.
Necessary information for sugars used in this study has been inserted. Lines 129-130.
2. Line 111. The teeth were immersed and sanitized in sodium hypochlorite (10% concentration).
How long the teeth immersed in sodium hypochlorite (10% concentration) until used? Sodium hypochlorite is a strong oxidizer, and oxidation reactions are corrosive. The teeth were chemically damaged.
Patricia/Josephine, can you add some info about the duration of the treatment? A reference to past work using the same treatment would be ideal. I don’t have that information.
As a necessary treatment of the specimen, the impact to the teeth was uniform among all tests thus no concern to the conclusions.
3. Line 129. 18 mM of glucose, GlcN, GlcNAc, or 9 mM of sucrose or lactose.
I think that the concentration of these addiction was 18mM for mono- and 9mM for disaccharide sugar. Does 18mM or 6 mM of saccharide sugar mimic the oral environment? Please provide related reference(s).
That is correct. Such concentrations are commonly used in similar studies to mimic the oral environment. A reference has been inserted. Line 131.
4. Line 149
Method of measurement of organic acids using LC-MS was not explained in Materials and methods, but in the caption in Figure 2. Please explain its detail in Materials and methods.
Necessary details of the LC-MS procedure have been added to the methods section. Lines 156-160.
5. Line 163. 2 mM of sucrose and 1% of additional carbohydrate as specified. Six samples of each specimen were kept in a 24-well plate, 3 in 1% of glucose, the other 3 in 1% of GlcNAc.
i) GlcN was not used in organic acids measurement and evaluation erode teeth and restorative surfaces. Are there any difference in the effect between GlcNAc and GlcN (if used)?
As explained in the manuscript (line 148), only glucose and GlcNAc were selected for metabolomics and erosion assays because GlcNAc showed the greatest difference in comparison to glucose in microbial composition study and in other previous research.
ii) Figure2 and 3 shows Glc and GlcNAc. Abbreviation of Glc is not defined in the manuscript.
Abbreviation of glucose has been inserted at two places, lines 129 and 252, for clarity.
6. Figure 1 indicated a metabolic shift in the microcosm biofilm caused by GlcNAc, but not by GlcN, Why?
Previous research on the impact of GlcN and GlcNAc to transcriptomes of model streptococci has demonstrated significant difference in their abilities to influence bacterial central metabolism (Chen et al, AEM, 2020), which may be the main reason behind this difference in microbial composition seen here. Specifically, catabolism of GlcNAc releases more acetate than GlcN due to one additional deacetylation reaction needed to degrade GlcNAc.
7. For Fig. 3 and 4, statistical analyzed methods were not fully described.
How about normal distribution and analysis of variance?
More information has been inserted in the legends of Fig. 3. No statistical treatment was performed for Fig. 4 partly because of the number of samples analyzed and partly because conclusions were drawn only from the derivatives of these data.
8. Line 194. Samples were sputter-coated with carbon to reduce charging effects during SEM analysis
If not EDS elemental analysis, carbon is not the best choice for coating. Gold or platinum is a better material than carbon.
We thank the reviewer for this comment.
9. Line 280. Figure 5. and Table 1.
i) The groups are placed in the order of tooth, titanium and ceramic in Figure 5, but in the order of tooth, ceramic and titanium in Table 1. Please arrange them consistently.
ii) In Figure 5, especially ceramic samples, scratches and streaks from polishing are visible. The use of filtering to remove these artifacts may be recommended. 
Adjustments have been made to Table 1 to improve consistency among the data.
Reviewer 2 Report
The manuscript titled “The effect of amino sugars on the composition and metabolism of a microcosm biofilm and the cariogenic potential against teeth and dental materials” reported the results on the effects of GlcNAc metabolism to the genomics and biochemistry of a saliva-derived biofilms. Authors demonstrated that GlcNAc was able to reduce the cariogenic potential of oral biofilms by modifying their biochemical and bacterial composition.
The results are interesting and clearly presented, therefore the manuscript can be considered suitable for the publication in the journal.
Only few issue need to be fixed:
- - In the introduction authors should add a brief discussion to clarify the key role of the bacterial biofilm formation in the development of antibiotic resistance. Recent references on the topic should be added: Current Medicinal Chemistry, 2022, 29, 4307–4310 and Future Medicinal Chemistry, 2021, 13, 529–531
- - Conclusion: this section shloud be implemented by discussing the meanings and the utility of the results described
- - References: check if Ref 27 and 28 are corrected
Author Response
We have carefully reviewed the comments. Our work has no direct relations with antibiotic resistance and references 27 and 28 are for two software programs. We implemented the conclusions with the suggested information (lines 367 to 369). Changes to the manuscript are shown in track changes.Reviewer 3 Report
In the manuscript authors present study in which they investigate the effects of GlcNAc metabolism to the genomics and biochemistry of a saliva-derived microbial community, and to the surface integrity of human teeth and restorative surfaces. The results support authors idea of the ability of amino sugars in reduction of the cariogenic potential of oral biofilms by altering their biochemistry and bacterial composition.Moreover it is shown that amino sugar metabolism appears to be less detrimental to teeth and restorative surfaces than glucose metabolism.
I find this paper interesting and support its publication.
Author Response
Thank you for the revision. All the changes in the manuscript are shown in track changes.